# Research on Insulator Defect Detection Based on an Improved MobilenetV1-YOLOv4

**DOI:** 10.3390/e24111588

**Published:** 2022-11-02

**Authors:** Shanyong Xu, Jicheng Deng, Yourui Huang, Liuyi Ling, Tao Han

**Affiliations:** 1School of Electrical & Information Engineering, Anhui University of Science and Technology, Huainan 232001, China; 2School of Electrical and Opto Electronic Engineering, West Anhui University, Lu’an 237012, China

**Keywords:** insulator defect detection, YOLOv4, Mobilenet-V1, scSE attention mechanism, depthwise separable convolution

## Abstract

Insulator devices are important for transmission lines, and defects such as insulator bursting and string loss affect the safety of transmission lines. In this study, we aim to investigate the problems of slow detection speed and low efficiency of traditional insulator defect detection algorithms, and to improve the accuracy of insulator fault identification and the convenience of daily work; therefore, we propose an insulator defect detection algorithm based on an improved MobilenetV1-YOLOv4. First, the backbone feature extraction network of YOLOv4 ‘Backbone’ is replaced with the lightweight module Mobilenet-V1. Second, the scSE attention mechanism is introduced in stages of preliminary feature extraction and enhanced feature extraction, sequentially. Finally, the depthwise separable convolution substitutes the 3 × 3 convolution of the enhanced feature extraction network to reduce the overall number of network parameters. The experimental results show that the weight of the improved algorithm is 57.9 MB, which is 62.6% less than that obtained by the MobilenetV1-YOLOv4 model; the average accuracy of insulator defect detection is improved by 0.26% and reaches 98.81%; and the detection speed reaches 190 frames per second with an increase of 37 frames per second.

## 1. Introduction

Insulators are unique insulating devices that are used on overhead transmission lines to protect the power transmission lines. There are mainly three types of insulators: ceramic, glass, and composite. Due to long-term exposure to the outdoors, short-circuit faults between lines are likely to occur when there are insulator problems such as bursting, string loss, and contamination, which result in significant harm to the normal power supply and service life of transmission lines. Therefore, insulator defect detection is important to ensure normal operation of power lines.

With the development of artificial intelligence and machine learning technology, many scholars have conducted research on target algorithms for insulator defect detection that can be divided into two main categories: One category is the two-stage target detection algorithm based on candidate regions and CNN extraction; the other category is the one-stage target detection algorithm that directly completes feature extraction, classification, and regression prediction [1]. The first category includes the RCNN [2], Fast RCNN [3], and Faster RCNN [4] two-stage target detection algorithms; the second category includes YOLO (You Only Look Once) [5], SSD (Single Shot Multibox Detector) [6], YOLOv2 [7], YOLOv3 [8], YOLOv4 [9], etc. The authors of [10] proposed an improved YOLOv5 insulator breakage detection algorithm; this algorithm improved the accuracy of the algorithm by adding the attention mechanism ECA-NET (efficient channel attention) and by using the Soft NMS algorithm to reallocate the original candidate frames. The detection effect of overlapping targets was strengthened, which could meet the engineering application. In a study by [11], an insulator defect detection method was proposed that adopted an RPN network for feature extraction to realize the detection of tiny defects on insulators. The advantage of this method was that it had high detection accuracy. However, it had low detection speed, i.e., only 12.8 frames per second and it could not be deployed on the mobile end. A study by [12] adopted ResNet as a backbone network, and equipped it with a batch of normalized convolutional attention modules (BN-CBAM) that could better utilize channel information and could enhance the ability of different channels to feature maps. In this case, this model had a large number of parameters. Mobile deployment requires more computing resources and memory. The authors of [13] proposed a defect detection algorithm deployed on FPGA. The algorithm had high efficiency, power loss was only 10 W, and hardware deployment was realized. Ref. [14] proposed an insulator fault detection algorithm based on a convolutional neural network, which, first, cascaded the detection network and the fault classification network to realize the position detection and fault classification of insulators, then detected faulty insulator(s) in images with complex background and high resolution, and finally, realized the fault detection. Unfortunately, there were few datasets and the identified faults were limited. Ref. [15] proposed a YOLOv5 insulator defect detection method; although it realized fast detection, this method sacrificed accuracy for speed, with only 81% accuracy, and therefore, could not be used in practical applications. In [16], the authors proposed an insulator detection method based on YOLOv2 that could complete the detection of 25 images per second, however, with an accuracy of only 88%, and therefore, may have missed detection. Ref. [17] proposed an insulator defect detection method based on YOLOv5s and designed a new residual module to reduce network parameters and to extract more useful features. The method had high detection speed but at the cost of network accuracy. Ref. [18] proposed an improved YOLOv3-based insulator detection with a new feature pyramid network, which had high detection accuracy for insulator defects. However, the network could not learn independently. Ref. [19] proposed an aerial image insulator detection based on an improved YOLOv3, which utilized multi-level feature mapping modules in the network. This model had better detection accuracy, but increased the complexity of the network. Ref. [20] proposed an insulator fault detection model based on an improved YOLOv3 with the utilization of an SPP network and a multi-scale prediction network. The performance of this method for insulator detection was relatively good, but the weight of the improved network model was up to 255 MB with significant use of memory. Ref. [21] proposed an insulator fault detection model based on an improved Faster RCNN that adopted a feature pyramid network to complete the insulator image location under a complex background and insulator identification. However, this method had high requirements for hardware. Although the detection accuracies of the proposals by [19,20,21] were improved, they were difficult to deploy to actual models and practical applications. In [22], an image texture segmentation algorithm was proposed for aviation insulators. However, the recognition effects were unsatisfactory as the texture features of the insulators were very similar to leaves, though different from most backgrounds. Ref. [23] proposed a multi-scale feature insulator detection algorithm that, first, extracted the local features of insulators, and then trained local features to obtain spatial order features to improve the robustness of the algorithm. However, this algorithm was highly complex, time-consuming, and incapable of satisfying real-time applications. Ref. [24] proposed a multi-scale residual neural network-based insulator surface damage identification method that could effectively detect insulators in single backgrounds while ignoring the identification of insulators in complex backgrounds. Ref. [25] proposed an improved U-Net insulator image segmentation method based on attention mechanism, which embeded the ECA-Net attention mechanism in the encoding stage of U-Net to improve the semantic feature extraction ability of the model and further the accuracy of the insulator image segmentation. Ref. [26] proposed an insulator self-explosion detection method based on YOLOv4 that, first, fused the shallow feature map into the feature pyramid, and then adopted the SENet structure to improve the recognition accuracy of network. This method had high accuracy and a slight decrease in detection speed. Ref. [27] proposed a new insulator defect detection algorithm using deep learning and morphological detection that adopted a residual network to extract the morphological features of insulators, and then the image segmentation pixel clustering method to establish a mathematical model of insulator defects. The detection accuracy of this method was good, but the network parameters were more complicated. Ref. [28] proposed an insulator string detection method based on an improved YOLOv5 that adopted an EIOU loss function and the AFK-MC2 anchor point generation method to detect insulators. Ref. [29] proposed an insulator defect detection method based on an improved lightweight YOLOv4, reaching a detection accuracy of 93.81% and a detection speed of 53 frames per second. Ref. [30] proposed an improved insulator detection algorithm based on YOLOvX that adopted an improved Siou loss function to speed up model convergence, and was embedded with the ECA attention mechanism. The research results showed that the detection accuracy of this method reached 97.18%, and the detection speed was 71 frames per second. Although both two-stage and one-stage target detection algorithms have achieved great success in insulator recognition, due to the limitations of storage space and power consumption, many scholars have begun to study a lightweight model with reduced complexity and improved detection speed that can be deployed on tiny devices. A traditional CNN has high memory and computational demands, which makes it unable to run on mobile and embedded devices. MobileNet [31,32,33] was proposed by the Google team, in 2017, with a focus on lightweight CNN networks in mobile terminals or embedded devices. It used depth convolution in the network to reduce the amount of computations and parameters. Using the reciprocal residual structure in the network could reduce the memory consumption during reasoning. Wang [34] proposed a CSPNet lightweight network, in 2019, that integrated gradient changes into a feature map from beginning to end, and thus, reduced the amount of computations and ensured accuracy. This method reduced the amount of calculation and improved the running speed of the model without reducing the accuracy of the model and could be used for mobile deployment. In 2020, Huawei proposed a new lightweight network, i.e., GhostNet [35]. We knew that the redundancy of the feature map was very important, therefore, we designed a Ghost module that realized the operation with fewer computations to generate the redundant feature maps, and deployed the network to a mobile terminal. In order to make the algorithm model smaller, we applied it to insulator detection.

To further improve insulator detection accuracy and speed, to reduce algorithm complexity, and to make the algorithm easy to be deployed on hardware equipment (such as drones), in this paper, we propose a lightweight insulator defect detection algorithm based on an improved MobilenetV1-YOLOv4 that reconstructs the backbone feature extraction network of YOLOv4 through the lightweight module Mobilenet-V1, introduces the scSE attention mechanism to enhance feature extraction, and adopts depthwise separable convolution to reduce the overall number of parameters in the network. The effectiveness of this proposed algorithm was verified through simulation and experimental analysis.

## 2. Fundamentals of MobilenetV1-YOLOv4

YOLOv4 is a high-accuracy, one-stage target detection algorithm that has been developed from YOLOv3. In 2020, Bochkovskiy et al. published the YOLOv4 model and explained it. The MobilenetV1-YOLOv4 algorithm is a lightweight network model obtained from YOLOv4. It consists of the following three parts: backbone feature extraction network, feature pyramid network (PANet, SPP), and classification regression layer (YOLO Head). Figure 1 shows the network structure of MobilenetV1-YOLOv4.

### 2.1. Backbone Feature Extraction Network

The backbone feature network of YOLOv4 is CSPDarkNet53. It adopts the CSPnet structure to split the original residual block into two parts: the main body and the branch. The main body is still a residual block, and the branch is a large residual edge that is connected to the main body through a small amount of processing. The benefit of this is that gradients do not disappear as the network depth complexity increases. To reduce network complexity, CSPDarkNet53 is replaced by MobilenetV1. The network structure adopts depthwise separable convolution that decomposes standard convolution into two steps, i.e., depthwise convolution and point-by-point convolution. This can significantly reduce the size of the algorithmic model. Finally, to avoid information loss of image features due to multiple convolutions, output is still in three dimensions, i.e., 13 × 13 × 1024, 26 × 26 × 512, and 52 × 52 × 256.

### 2.2. Feature Pyramid Network

A feature pyramid network enhances feature extraction for three dimensions of the initial feature network output, which can be divided into two main parts: SPP and PANet. The result obtained by the convolution of the last feature layer of the MobilenetV1 network is subjected to SPP max pooling, in which there are four different compositions of convolutional kernels; the pooling kernel sizes for max pooling are 13 × 13, 9 × 9, 5 × 5, and 1 × 1, respectively. Finally, the splicing of the quantity of channels is realized and sent to the PANet network. The PANet structure is an instance segmentation algorithm that can be used for both top-to-bottom and bottom-to-top feature extractions. The size of the feature map is changed through upsampling and downsampling to realize dimension splicing and complete repeated feature extraction.

### 2.3. Classification and Regression Layer

In the PANet network, 3 × 3 convolution is first used to convolve the result of the feature pyramid, and then the 1 × 1 convolution, and finally three prediction results of different sizes are obtained. However, the positions of the prediction results and the final prediction frame do not correspond, and therefore, decoding is demanded to finally acquire the prediction information.

## 3. Improved MobilenetV1-YOLOv4 Algorithm

In this paper, the YOLOv4 algorithm is adopted as the basic network. The original CSPdarknet53 network is replaced with a lighter network Mobilenet-V1 to make the detection speed of the model faster. The 3 × 3 ordinary convolutions in the PANet network and SPP are replaced by depthwise separable convolution to further reduce the number of network parameters. To maintain network accuracy, the scSE attention mechanism is added to the three dimensions of the preliminary feature extraction, and also to the results of upsampling in the enhanced feature extraction network. Finally, the convolution layer after SPP is modified to five layers. The improved network structure is shown in Figure 2.

As shown in Figure 2, the convolution size in the CBR module is 1 × 1, and later, normalization is used to avoid the disappearance of gradient and to speed up the convergence; the activation function can effectively increase the nonlinearity of the network and make it learn better; the DBRCBR module contains two types of convolutions, i.e., depthwise convolution and point convolution, to form depthwise separable convolution. Therefore, the traditional 3 × 3 convolution is replaced with this depthwise separable convolution. Conv2D in the DBRCBR module is point convolution, and it is actually a 1 × 1 convolution. Its function is to freely change the number of output channels. Next, it performs channel fusion on the feature map of depth convolution output. After the convolution operations of depth convolution and point convolution, this is the operation process of depth separable convolution. The scSE attention mechanism can enhance the feature information that is needed and can suppress the feature information that is not needed.

### 3.1. scSE Attention Mechanism

The spatial and channel squeeze and channel excitation (scSE) attention mechanism module is a combination of two modules, spatial squeeze and channel excitation (cSE) and channel squeeze and spatial excitation (sSE).

The cSE attention module is shown in Figure 3. First, global pooling is performed on the feature map to compress the feature image information. It can generate a Z vector (dimension is 1 × 1 × C) whose dimension changes from [C, H, W] to [C, 1, 1], and compresses the height and width of the feature image to 1 × 1. Then, two fully connected layers are used to process feature information; the fully connected neuron quantity is less than the input feature layer the first time, while it is equal the second time. After two times of full connection, the C-dimension vector is obtained, and the Sigmoid activation function is used for normalization to fix its value between zero and one. At this time, the weight of each channel of the input feature layer is known. Finally, multiply the obtained weights with the original feature map to obtain the calibrated results, which improves the ability of the network to extract channel features.

Figure 4 displays the sSE attention mechanism module whose role is to squeeze the feature map along the channel and spatially motivate. First, operate on the input feature map. The channel compression is performed using a 1 × 1 × 1 convolutional method, which changes the feature map from the initial [C, W, H] to [1, H, W]. Then, a new spatial feature map is obtained by the Sigmoid activation function method, and finally the obtained new spatial feature map is multiplied with the original feature map to achieve spatial information calibration. This operation has a significant effect on the spatial position of related features, thereby, omitting irrelevant features and improving the learning ability of spatial feature information.

The scSE attention mechanism module adds the new features obtained by both cSE and sSE attention modules to complete the stacking of information. The scSE attention mechanism can enhance the channel and spatial information, and can improve the detection accuracy of the whole network, as shown in Figure 5.

### 3.2. Depth Separable Convolutional Module

Depthwise separable convolution is a combination of depthwise convolution and point convolution. It can reduce the number of parameters calculated, and can reduce the model size. The process of depthwise separable convolution is shown in Figure 6. First, the first convolutional operation is performed on the input RGB image, and each convolutional kernel is responsible for calculating one channel. The quantity of convolution kernels is similar to that of the channels in the previous-layer feature map. Through this operation, three characteristic graphs are obtained. However, the number of feature maps after depth convolution is consistent with the number of channels in the input layer, so more and more effective feature maps cannot be obtained. Meanwhile, this operation also requires a separate convolutional computation for each channel of the input layer, which cannot effectively utilize the feature information of different channels at the same spatial position. Thus, point convolution is adopted to convolve these feature maps again to generate new feature maps. The computation mode of point convolution is very similar to that of conventional convolution. The size of the convolution kernel of point convolution is 1 × 1 × C, where C represents the quantity of channels after depthwise convolution. Each convolutional kernel has three channels. Then, multiply and accumulate the convolutional kernel with the three channels of the input image to obtain new feature maps. The amount of these new feature maps is consistent with that of the convolutional kernels. After the above depthwise and point convolutions, we can finally acquire new features. In addition, the number of network parameters is significantly reduced, and the detection speed is quickened.

For ordinary convolution, the calculation formula of its parameter quantity is Formula (1) as follows:(1)$$N_{pc}=D_{F}\times D_{F}\times C_{i}\times C_{o}$$
where *D_F_* refers to the dimension of convolutional kernel, *C_i_* refers to the quantity of input channels, *C_o_* refers to the dimension of output, and *N_pc_* refers to the quantity of parameters.

The formula of network computation amount is Formula (2) as follows:(2)$$N_{cc}=H_{o}\times W_{o}\times C_{o}\times D_{F}\times D_{F}\times C_{i}$$
where *H_o_* and *W_o_* stand for the height and width of the input feature map, respectively, and *N_cc_* stands for the network computation amount.

For depthwise separable convolution, its parameter quantity is calculated by adding the total parameters obtained by depthwise convolution and point convolution. Formula (3) can be used to calculate the parameter quantity of depthwise convolution as follows:(3)$$N_{p1}=D_{F}\times D_{F}\times C_{i}$$

Formula (4) can be used to calculate its network computation amount as follows:(4)$$N_{c1}=H_{o}\times W_{o}\times C_{i}\times D_{F}\times D_{F}$$

Formula (5) can be used to calculate the parameter quantity of point convolution as follows:(5)$$N_{p2}=C_{i}\times C_{o}$$

Formula (6) can be used to calculate its network computation amount as follows:(6)$$N_{c2}=H_{o}\times W_{o}\times C_{o}\times 1\times 1\times C_{i}$$

Thus, the total parameters for depthwise separable convolution can be obtained by Formula (7):(7)$$N_{pd}=N_{p1}+N_{p2}$$

The total network computation amount can be obtained by Formula (8):(8)$$N_{cd}=N_{c1}+N_{c2}$$

Further, the ratio of the parameter quantity of depthwise separable convolution to that of ordinary convolution can be obtained as per Formula (9), from which we can know the number of parameters under depthwise separable convolution is $\frac{1}{C_{o}}+\frac{1}{D_{F}^{2}}$ of the original one:(9)$$\frac{N_{pd}}{N_{pc}}=\frac{D_{F}\times D_{F}\times C_{i}+C_{i}\times C_{o}}{D_{F}\times D_{F}\times C_{i}\times C_{o}}=\frac{1}{C_{o}}+\frac{1}{D_{F}^{2}}$$

In the improved MobilenetV1-YOLOv4 algorithm network, the ordinary 3 × 3 convolutions in the PANet network and the SPP network are replaced by the depthwise separable convolution, and the weight after training is 57.9 MB. As compared with the weight of 244 MB trained by the YOLOv4 network, the size of the weight is reduced by 76.3%.

### 3.3. Mobilenet-V1 Network

In this paper, the original backbone network CSPdarknet53 is replaced by the Mobilenet-V1 lightweight network. The Mobilenet model is a lightweight network proposed by Google for devices with low computing power such as embedded devices. Its core operation is to form the main network by using depthwise separable convolutional blocks. Three preliminary effective features can be obtained through the Mobilenet-V1 network, which are 52 × 52 × 256, 26 × 26 × 512, and 13 × 13 × 1024. Table 1 demonstrates the Mobilenet-V1 structure, in which the ordinary convolution is represented by Conv, the depthwise separable convolution is represented by Conv dw, and the step size in the convolutional process is represented by S. Three layer structures of the original Mobilenet-V1 are removed, namely global average pooling, fully connected layer, and Softmax.

## 4. Experimental Platform Construction and Training

### 4.1. Experimental Platform

This experiment is completed under Windows11 system with a computer configuration as follows: the 12th Gen Intel(R) Core(TM) i7-12700KF@3.61 GHz CPU, 32 GB memory, NVDIA GeForce RTX 3080 Ti GPU, 12GB video memory, software Anaconda3, version 3.7 Python, and Tensorflow2.5 deep learning framework.

### 4.2. Data Collection and Processing

The dataset of this paper comes from the Baidu open source dataset, including 4147 pictures of diversified and defective insulators, and various types of insulator defect data. In order to enable the network model to learn the characteristics and location information of insulators and to improve the accuracy of model output, insulator defects in the image can be marked. The .xml label files are formed by using the labelimg software to mark insulator defects. The boundary box is minimized as far as possible during the image marking of insulator defects to reduce the influence of background. The annotation results are saved in PASCAL VOC format, and the generated XML format file is saved in the pre-created folder. The ratio of training set plus validation set to test set is 9:1, and the ratio of training set to validation set is 9:1. Some dataset images are shown in Figure 7.

### 4.3. Model Training

A total of four groups of comparative experiments were performed in this study. The first group contains YOLOv4, MobilenetV1-YOLOv4, and improved MobilenetV1-YOLOv4; the second group is the comparison with other lightweight models, namely MobilenetV1-YOLOv4, MobilenetV2-YOLOv4, MobilenetV3-YOLOv4, and Ghostnet-YOLOv4; the third group is the comparison with improved networks, respectively, improved MobilenetV1-YOLOv4, improved MobilenetV2-YOLOv4, improved MobilenetV3-YOLOv4, and improved Ghostnet-YOLOv4. The fourth group of experiments was conducted in a deeply separable convolutional network by adding the scSE attention mechanism. It is compared with the current algorithm YOLOv5.

The idea of migration learning is applied in the training of the network, and the pretraining weights are obtained by using the training VOC dataset. During the training process, each network is trained for a total of 250 times. For the first 50 times of training, the backbone network of the model is frozen by the frozen training to improve the training efficiency and avoid the destruction of weights. In addition, the occupied video memory is also small in the process. The batch size is set to 32 during frozen training. Afterwards, unfrozen training is performed to unfreeze the backbone, which increases the occupation of the network video memory. At this time, the batch size is set to 16. By judging the current batch size, the learning rate of the model can be adaptively adjusted from the minimum 0.0001 to the maximum 0.01. Three different YOLO Heads are acquired by improving the training of the network, and finally the prediction frame is obtained by using the non-maximum suppression method. The essence of the non-maximum suppression method is to search local maxima and suppress nonmax elements. First, set the confidence threshold of the target box. The threshold set in this study is 0.5. Then, arrange the list of candidate boxes in descending order according to the confidence. In the identified target categories, select and retain the bounding box A with the highest confidence, and then calculate the IoU of bounding box A and the remaining boxes B. If the IoU value is greater than the threshold, remove B. Repeat this step until the iteration of a target class is completed and the required target box is finally output.

In the training process, the mosaic data enhancement method is used, and four pictures are randomly read at one time from the dataset for scaling or cropping to get a new picture after combination. The new image has a different background, which enriches the spatial semantic information and enhances the generalization ability of the model. By using the label smoothing training method, the data can be better calibrated and can be more accurately predicted. The cosine annealing algorithm is used to avoid the network falling into the local optimal solution. Formula (10) illustrates the principle of using the cosine annealing algorithm to prevent the network from falling into the local optimal solution:(10)$$\eta_{t}=\eta_{\min}^{i}+\frac{1}{2}\left(\eta_{\max}^{i}-\eta_{\min}^{i}\right)\left(1+\cos\left(\frac{T_{cur}}{T_{i}}\pi \right)\right)$$
where *i* represents the ordinal of the index value; $\eta_{\min}^{i}$ and $\eta_{\max}^{i}$ stands for the minimum and maximum values of learning rate, respectively; *T_cur_* refers to the current number of cycles; and *T_i_* means the total number of cycles under the current operating environment.

### 4.4. Loss Function

The loss function of the improved MobilenetV1-YOLOv4 consists of three parts, which are, in turn, position regression loss, object confidence loss, and classification loss, as shown in Formula (11):(11)$$Loss=Loss_{loc}+Loss_{obj}+Loss_{cls}$$

Formula (12) shows the position regression loss:(12)$$Loss_{loc}=\lambda_{coord}\sum_{i=0}^{s\times s}\sum_{j=0}^{M}I_{ij}^{obj}\left(2-W_{i}\times h_{i}\right)\left(1-CIOU\right)$$
where the computations for *CIOU*, *β*, and *v* follow Formulas (13)–(15), respectively:(13)$$CIOU=IOU-\frac{\rho^{2}\left(b,b^{gt}\right)}{c^{2}}-\beta \nu$$
(14)$$\beta =\frac{\nu }{1-IOU+\nu }$$
(15)$$\nu =\frac{4}{\pi^{2}}{\left(\arctan\frac{w^{gt}}{h^{gt}}-\arctan\frac{w}{h}\right)}^{2}$$

Formula (16) displays the object confidence loss:(16)$$\begin{array}{cc} Loss_{obj} & =-\sum_{i=0}^{s\times s}\sum_{j=0}^{M}I_{ij}^{obj}\left[\overset{⌢}{C_{i}}\log\left(C_{i}\right)+\left(1-\overset{⌢}{C_{i}}\right)\log\left(1-\overset{⌢}{C_{i}}\right)\right]-\\ & \lambda_{noobj}\sum_{i=0}^{s\times s}\sum_{j=0}^{M}I_{ij}^{noobj}\left[\overset{⌢}{C_{i}}\log\left(C_{i}\right)+\left(1-\overset{⌢}{C_{i}}\right)\log\left(1-\overset{⌢}{C_{i}}\right)\right] \end{array}$$

Formula (17) displays the classification loss function:(17)$$Loss_{cls}=-\sum_{i=0}^{s\times s}\sum_{j=0}^{M}I_{ij}^{obj}\sum_{c\in classes}\left[\overset{⌢}{P_{i}}\left(c\right)\log\left(P_{i}\left(c\right)\right)+\left(1-\overset{⌢}{P_{i}}\left(c\right)\right)\log\left(1-P_{i}\left(c\right)\right)\right]$$

From (12) to (17), *λ_coord_* and *λ_noobj_* represent the weight coefficient; s × s represents the grid number; *M* represents each anchor box; *w_i_* and *h_i_*, respectively, represent the width and height of the prediction box center point; *IOU* represents the ratio of the predicted value to the true value; $\rho^{2}\left(b,b^{gt}\right)$ represents the Euclidean distance between the center points of prediction box and true value; *C* represents the diagonal distance of the smallest closure region that can contain both the prediction box and the true value; *β* is a parameter to measure the consistency of length-width ratio; *V* is a trade-off parameter; $\overset{⌢}{C_{i}}$ is the predicted confidence degree and *C_i_* is the actual confidence degree; $I_{ij}^{obj}$ refers to the jth prediction box of the ith grid that is responsible for predicting the target; and $I_{ij}^{noobj}$ refers to the jth prediction box of the ith grid that contains no prediction of target. Figure 8 demonstrates the improved MobilenetV1-YOLOv4 loss function. The smaller the loss function, the better the robustness of the model.

As shown in Figure 8, the loss function of the training set and the verification set has a significant downward trend. After 200 times of training, the loss function gradually converges to a fixed value. After reaching 250 times, the loss function is basically unchanged. This indicates that the improved network has high convergence speed and the trainings work well.

## 5. Experimental Results and Discussion

This paper takes precision *P*, recall rate *R*, average precision (*mAP*), and frame rate (*FPS*) as the main evaluation indicators of the proposed algorithm in the experiment. Their formulae are, respectively, as follows:(18)$$P=\frac{TP}{TP+FP}$$
(19)$$R=\frac{TP}{TP+FN}$$
(20)$$AP=\int_{0}^{1}P(R)dR$$
(21)$$mAP=\frac{1}{n}\sum_{i=1}^{n}AP$$
(22)$$f_{FPS}=\frac{N}{t}$$

In these formulae, *TP* refers to the number of correctly predicted positive samples, *FP* the number of incorrectly predicted positive samples, and *FN* the number of incorrectly predicted negative samples; *n* represents the number of object detection categories, *N* represents the number of detected pictures, and *t* represents the detection time. An intersection ratio, which is the ratio of the intersection of the predicted frame and the real frame to their union, is introduced and set to 0.5 in this experiment. Table 2 provides the experimental results of the first group containing YOLOv4, MobilenetV1-YOLOv4, and the improved MobilenetV1-YOLOv4.

As can be seen from Table 2, the recall rate of the improved MobilenetV1-YOLOv4 is 21.7% higher than that of YOLOv4, and it reaches a detection speed of 190 frames/s, which may be due to the addition of the scSE attention mechanism and the depthwise separable convolution. In addition, as compared with the YOLOv4 algorithm, the detection speed of the improved MobilenetV1-YOLOv4 also increases by 2.6 times, and the detection precision improves by 4.14%; as compared with MobilenetV1-YOLOv4, the improved MobilenetV1-YOLOv4 also has a higher detection speed and a higher detection precision; the model weight of the improved MobilenetV1-YOLOv4 is significantly reduced as compared with YOLOv4 and MobilenentV1-YOLOv4. This is because scSE can automatically learn the effective features of an image from both image space and feature channel, can suppress useless redundant features, and can better retain image edge information. Finally, more useful features are obtained through feature splicing and convolution. The improved algorithm can identify insulator defects better.

Table 3 presents the comparisons among the improved MobilenetV1-YOLOv4, the MobilenetV1-YOLOv4, and the lightweight algorithms MobilnetV2-YOLOv4, MobilnetV3-YOLOv4, and Ghostnet-YOLOv4.

As can be seen from Table 3, the recall rate of the improved MobilenetV1-YOLOv4 is higher than that of other lightweight algorithms, and its mAP value is higher than that of other lightweight algorithms except that of MobilenetV2-YOLOv4. In addition, the detection speed of the improved MobilenetV1-YOLOv4 is optimal. Although MobilenetV1-YOLOv4, MobilenetV2-YOLOv4, MobilentV3-YOLOv4, and Ghostnet-YOLOv4 are lightweight networks, among the enhanced feature extraction networks, 3 × 3 convolution is still an ordinary convolution. The large number of network parameters leads to slower detection speed and greater model weight. The improved network is basically composed of deeply separable convolution and 1 × 1 convolution composition. This convolution can improve the real-time performance of the algorithm, and under the action of the scSE attention mechanism, it can ensure that the network accuracy will not decline.

Table 4 shows the third group of comparative experiments, which all have the added scSE attention mechanism. Finally, there are five convolution layers behind the SPP, and the 3 × 3 convolution is replaced by a depthwise separable convolution.

As can be seen from Table 4, as compared with other improved lightweight networks, the improved MobilenetV1-YOLOv4 still has the best detection accuracy and detection speed. In Table 4, all the lightweight networks have been improved. Because the improved Ghostnet-YOLOv4, improved MobilenetV2-YOLOv4, and improved MobilenetV3-YOLOv4 have a small number of network model parameters, the accuracy is lower than that of the improved MobilenetV1-YOLOv4. Through experimental verification, the detection speed of the improved MobilenetV1-YOLOv4 is optimal, with detection accuracy of 98.81% and detection speed of 190 frames/s, therefore, achieving significant improvement in its detection of insulator defects.

As can be seen from Table 5, first, scSE attention mechanism (2 scSEs) is only added after the upsampling results. Secondly, scSE (3 scSEs) is added after preliminary feature extraction. In this paper, we argue that embedding scSE attention modules into different parts of the network produces different experimental results. The semantic information of the feature map initially extracted is not rich, but it still retains the medium and shallow texture information and contour information of the target in the feature map. This information is very important for target detection. After the initial extraction of the three dimensions, embedding the scSE attention module can better enhance the spatial features and channel features of the target in the feature map. In the PANet structure, its feature map shows richer semantic features, larger receptive field, and smaller feature map scale. The scSE attention module can no longer effectively distinguish important spatial and channel features from the highly fused small-scale feature map.

In the experiment, with a gradual increase in scSE, the accuracy gradually increases, but the detection speed becomes smaller and smaller, because, in the same network, the network parameters are also increased. Finally, as compared with YOLOv5, although the accuracy is reduced, the detection speed and model weight are still better than YOLOv5.

Figure 9a–d show the P-R curves of the above four groups of experiments. The insulator defect label name is 1.

In a P-R curve, P represents precision and R represents recall. With recall as the abscissa axis and precision as the ordinate axis, a P-R curve can intuitively display the overall accuracy and recall of the classification algorithm. We can illustrate the superiority of the algorithm by comparing the area size under the P-R curve. It can be seen from Figure 9a–c that the area of the improved MobilenetV1-YOLOv4 curve is large, indicating that its detection accuracy is good. The green line represents YOLOv5, which occupies the largest area due to its highest accuracy.

Figure 10 is a chart of insulator defect prediction results. Regardless of whether the insulator defects in the figure are large or small, they can be effectively detected without missed detection. In addition, in each picture, the confidence level of the improved algorithm is higher than YOLOv4, indicating that the detection accuracy of this algorithm is higher. When the defect is at the edge of the image, the improved YOLOv4 detection effect is better.

## 6. Conclusions

In this paper, we put forward an insulator defect detection algorithm based on an improved MobilenetV1-YOLOv4. Comparative experiments with other algorithms prove the superiority of the proposed algorithm model, which has good detection accuracy while ensuring the speed.

(1) The accuracy of the insulator defect detection algorithm based on an improved MobilenetV1-YOLOv4 is up to 98.81% with a detection speed of 190 frames/s, which is 116 frames/s, 37 frames/s, 51 frames/s, 61 frames/s, and 72 frames/s faster than that of YOLOv4, MobilenetV1-YOLOv4, MobilenetV2-YOLOv4, MobilenetV3-YOLOv4, and Ghostnet-YOLOv4, respectively. As compared with the original YOLOv4, the detection accuracy of the proposed insulator defect detection algorithm is 4.14% higher.

(2) We obtain a brand new algorithm structure by replacing the backbone network, adding the scSE attention mechanism, and using depthwise separable convolution to reduce the quantity of network parameters. The experimental results show that the proposed algorithm can effectively detect insulator defects with highly improved detection accuracy and speed.

(3) We put forward a lightweight detection algorithm that is significantly advantageous for real-time detection of insulator defects. This is a direction worthy of research. In the future, we plan to further research insulator faults such as bursting and contamination. With the emergence of a new target detection algorithm, the algorithm proposed in this paper needs to be further optimized. At present, it will take some time to complete the hardware deployment. In the future, we plan to consider applying this model to other detections to achieve the real-time performance and generalization ability of this algorithm.

## Figures and Tables

**Figure 1 entropy-24-01588-f001:**
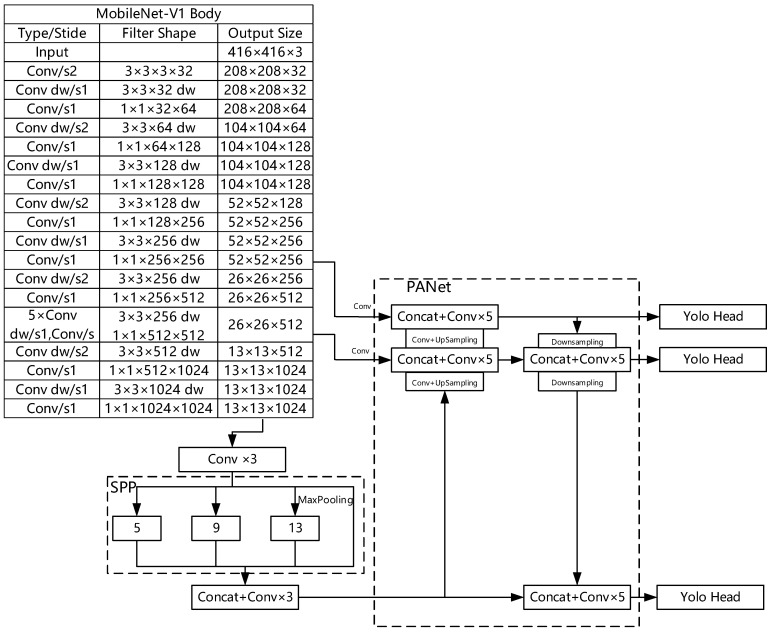
The network structure of MobilenetV1-YOLOv4.

**Figure 2 entropy-24-01588-f002:**
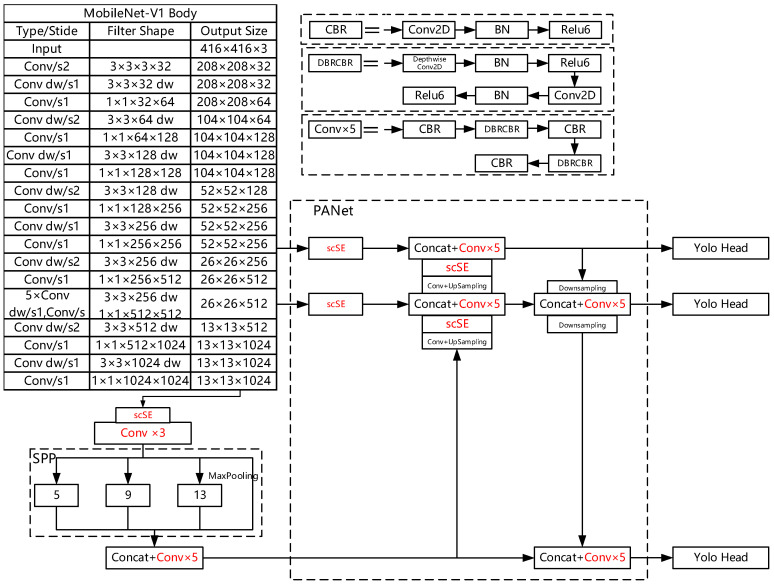
Improved MobilenetV1-YOLOv4 network.

**Figure 3 entropy-24-01588-f003:**
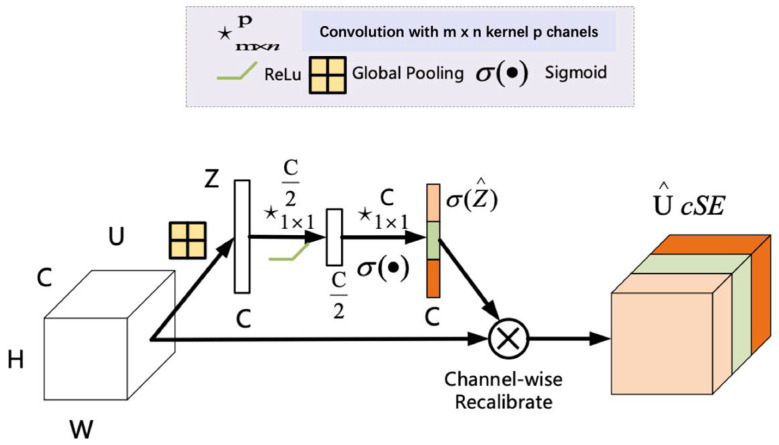
Spatial squeeze and channel excitation (cSE).

**Figure 4 entropy-24-01588-f004:**
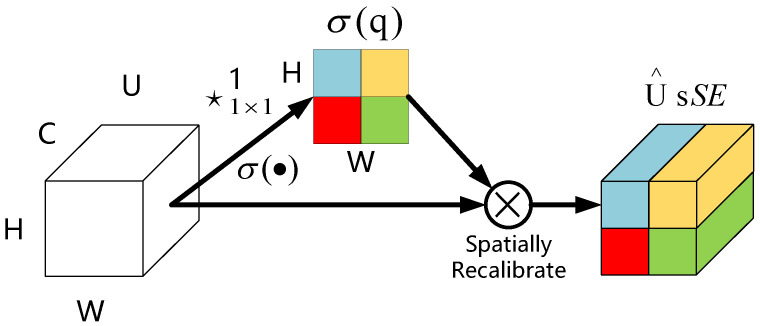
Channel squeeze and spatial excitation (sSE).

**Figure 5 entropy-24-01588-f005:**
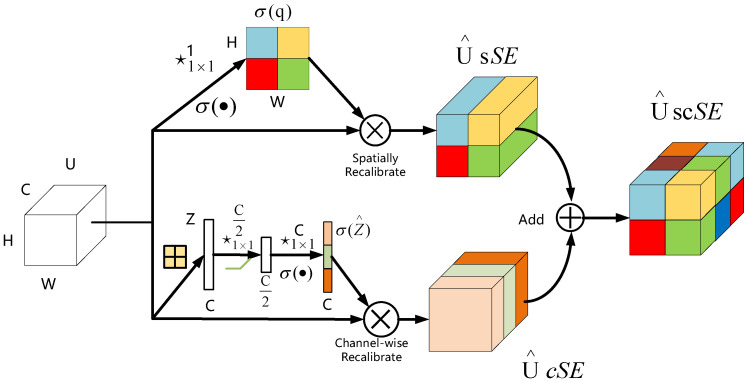
Concurrent spatial and channel squeeze and channel excitation (scSE).

**Figure 6 entropy-24-01588-f006:**
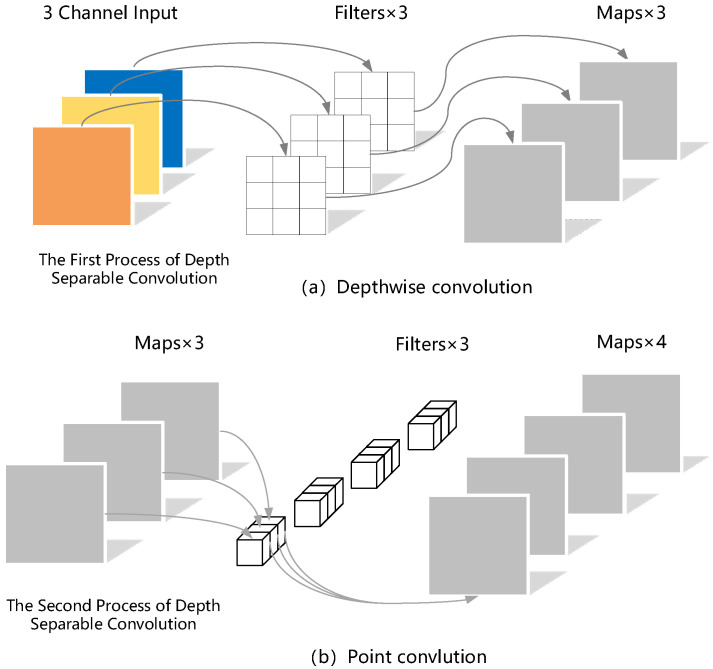
Depthwise separable convolution.

**Figure 7 entropy-24-01588-f007:**
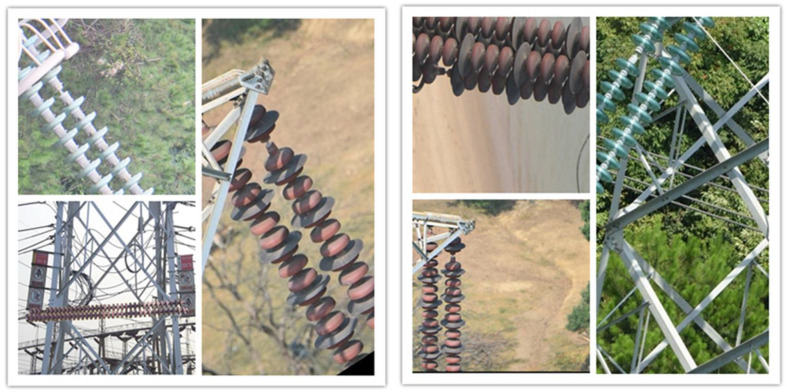
Several sample images.

**Figure 8 entropy-24-01588-f008:**
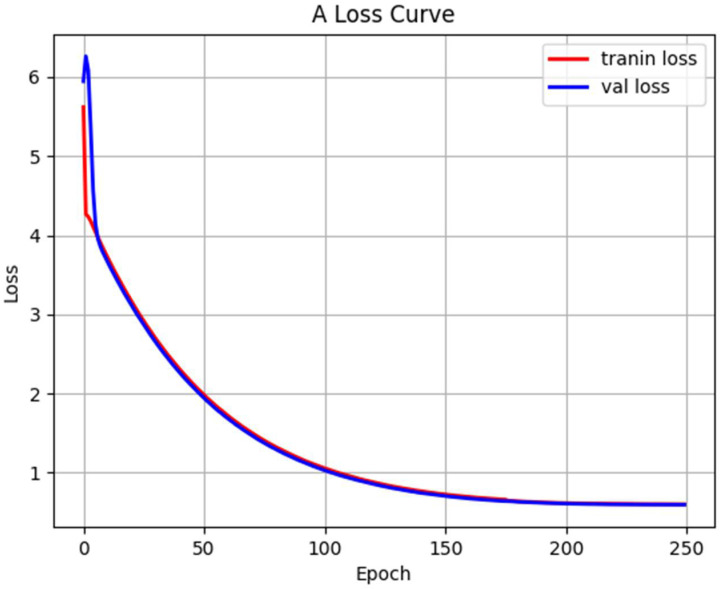
Loss function.

**Figure 9 entropy-24-01588-f009:**
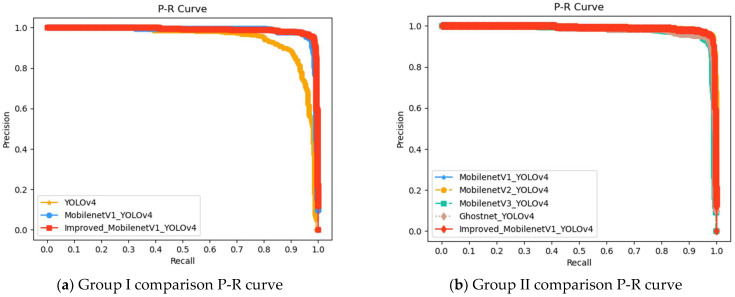
P-R curves based on YOLOv4.

**Figure 10 entropy-24-01588-f010:**
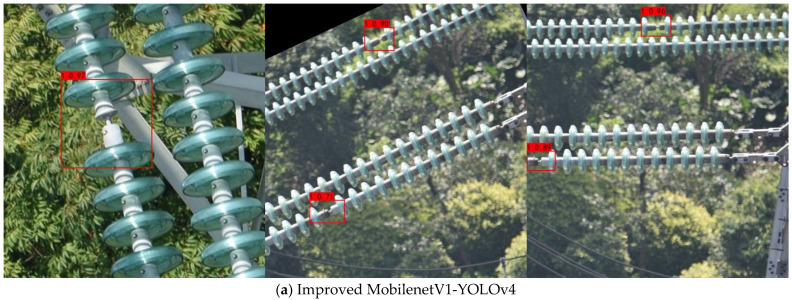
Insulator defect prediction result graph.

**Table 1 entropy-24-01588-t001:** Parameters of the Mobilenet-V1 network.

MobileNet-V1 Body		
Type/Stide	Filter Shape	Output Size
Input		416 × 416 × 3
Conv/s2	3 × 3 × 3 × 32	208 × 208 × 32
Conv dw/s1	3 × 3 × 32 dw	208 × 208 × 32
Conv/s1	1 × 1 × 32 × 64	208 × 208 × 64
Conv dw/s2	3 × 3 × 64 dw	104 × 104 × 64
Conv/s1	1 × 1 × 64 × 128	104 × 104 × 128
Conv dw/s1	3 × 3 × 128 dw	104 × 104 × 128
Conv/s1	1 × 1 × 128 × 128	104 × 104 × 128
Conv dw/s2	3 × 3 × 128 dw	52 × 52 × 128
Conv/s1	1 × 1 × 128 × 256	52 × 52 × 256
Conv dw/s1	3 × 3 × 256 dw	52 × 52 × 256
Conv/s1	1 × 1 × 256 × 256	52 × 52 × 256
Conv dw/s2	3 × 3 × 256 dw	26 × 26 × 256
Conv/s1	1 × 1 × 256 × 512	26 × 26 × 512
5 × Conv dw/s1, Conv/s1	3 × 3 × 256 dw, 1 × 1 × 512 × 512	26 × 26 × 512
Conv dw/s2	3 × 3 × 512 dw	13 × 13 × 512
Conv/s1	1 × 1 × 512 × 1024	13 × 13 × 1024
Conv dw/s1	3 × 3 × 1024 dw	13 × 13 × 1024
Conv/s1	1 × 1 × 1024 × 1024	13 × 13 × 1024

**Table 2 entropy-24-01588-t002:** Original contrast experiment (the first group).

Algorithm	Recall Rate (R)/%	mAP/%	FPS Frame/s	Model Weight/MB
YOLOv4	77.59	94.41	74	244
MobilenetV1-YOLOv4	92.29	98.55	153	155
Improved MobilenetV1-YOLOv4	93.25	98.81	190	57.9

**Table 3 entropy-24-01588-t003:** Comparison experiment of mainstream lightweight networks (the second group).

Algorithm	Recall Rate (R)/%	mAP/%	FPS Frame/s	Model Weight/MB
MobilenetV1-YOLOv4	92.29	98.55	153	155
MobilenetV2-YOLOv4	92.53	98.95	139	148
MobilenetV3-YOLOv4	90.12	97.74	129	152
Ghostnet-YOLOv4	88.92	98.02	118	150
Improved MobilenetV1-YOLOv4	93.25	98.81	190	57.9

**Table 4 entropy-24-01588-t004:** Comparative experiment after network improvement (the third group).

Algorithm	Recall Rate (R)/%	mAP/%	FPS Frame/s	Model Weight/MB
Improved Ghostnet-YOLOv4	91.33	97.98	135	43.8
Improved MobilenetV2-YOLOv4	92.77	98.13	168	45.5
Improved MobilenetV3-YOLOv4	90.84	97.98	164	48.9
Improved MobilenetV1-YOLOv4	93.25	98.81	190	57.9

**Table 5 entropy-24-01588-t005:** Comparative experiments (the four group).

Algorithm	scSE	Recall Rate (R)/%	mAP/%	FPS Frame/s	Model Weight/MB
Improved Mobi-lenetV1-YOLOv4	2	90.84	98.07	206	52
Improved Mobi-lenetV1-YOLOv4	3	90.84	98.18	196	57.6
Improved Mobi-lenetV1-YOLOv4	5	93.25	98.81	190	57.9
YOLOv5		98.55	99.67	109	81.8

## Data Availability

The data that support the findings of this study are available on request from corresponding author.

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
