# Peer review of "Research on Insulator Defect Detection Based on an Improved MobilenetV1-YOLOv4"

_entropy, 2022, doi:10.3390/e24111588_

Round 1

Reviewer 1 Report

(1) In section 3 of this paper, the author mentions "the DBRCBR module contains two kinds of convolutions, depthwise convolution and point convolution, to form depthwise separable convolution", while the "DBRCBR" described in Figure 2 only contains "depthwise convolution" and does not involve " depthwise separable convolution".

(2) In section 4.4 of this paper, the author uses non-maximum suppression algorithm to obtain the prediction frame, but the parameter settings related to the non-maximum suppression algorithm are not given and need to be supplemented.

(3) In this paper, the scSE attention mechanism is added to the PANet network, and it is best to explain the impact of its added position and number on the model detection results, and provide a theoretical basis.

(4) This paper proposes an insulator defect detection method based on the improved MobilenetV1-YOLOv4, which introduces the MobilenetV1 backbone feature extraction network, depthwise separable convolution and scSE attention mechanism, thus to greatly reduces the number of parameters, but the mAP value is increased by 4.4 %, whether the model has sufficient generalization ability for other related datasets.

(5) In section 5 of this paper, the author provides different models of P-R curves, but no relevant comparative analysis is carried out, which needs to be supplemented.

(6) In section 5 of this paper, the author gives the results of the improved MobilenetV1-YOLOv4 detection of insulators, and does not give the detection results of other related models. It is recommended to supplement different models to predict the comparison results of the same test image.

(7) Some image samples in the data set are given in Figure 7 in Section 4.2 of this paper, but they are all single images with a single insulator defect. Whether there are multiple insulator defects in a single image, which needs to be supplemented.

(8) Section 4.2 of the article divides the image data set into training set, test set and validation set, but Figure 8 in Section 4.4 does not show the loss curve for validation, which needs to be supplemented.

Author Response

Response to Reviewer 1 Comments

Point 1: In section 3 of this paper, the author mentions "the DBRCBR module contains two kinds of convolutions, depthwise convolution and point convolution, to form depthwise separable convolution", while the "DBRCBR" described in Figure 2 only contains "depthwise convolution" and does not involve " depthwise separable convolution".

Response 1: Thanks for your comment. According to your suggestion, We added the point convolution and depth separable convolution on the fifth page of the paper

Point 2: In section 4.4 of this paper, the author uses non-maximum suppression algorithm to obtain the prediction frame, but the parameter settings related to the non-maximum suppression algorithm are not given and need to be supplemented.

Response 2: Thanks for your suggestion. As you suggested, we added the specific content of the non maximum suppression algorithm on page 11 of the paper, and gave the parameter settings

Point 3: In this paper, the scSE attention mechanism is added to the PANet network, and it is best to explain the impact of its added position and number on the model detection results, and provide a theoretical basis.

Response 3: Thanks for your suggestion. As you suggested, we added the scsE attention mechanism on page 15 of the paper. The location and quantity of the added scsE attention mechanism have an impact on the model detection results, and gave the parameter settings

Point 4: This paper proposes an insulator defect detection method based on the improved MobilenetV1-YOLOv4, which introduces the MobilenetV1 backbone feature extraction network, depthwise separable convolution and scSE attention mechanism, thus to greatly reduces the number of parameters, but the mAP value is increased by 4.4 %, whether the model has sufficient generalization ability for other related datasets.

Response 4: Thanks for your comment. We have added relevant content on page 18 of the paper as you suggested.

Point 5: In section 5 of this paper, the author provides different models of P-R curves, but no relevant comparative analysis is carried out, which needs to be supplemented.

Response 5: Thanks for your comment. We added relevant contents of P-R curve on page 16 of the paper as you suggested.

Point 6: In section 5 of this paper, the author gives the results of the improved MobilenetV1-YOLOv4 detection of insulators, and does not give the detection results of other related models. It is recommended to supplement different models to predict the comparison results of the same test image.

Response 6: Thanks for your suggestion. We modified Figure 10 as you suggested.

Point 7: Some image samples in the data set are given in Figure 7 in Section 4.2 of this paper, but they are all single images with a single insulator defect. Whether there are multiple insulator defects in a single image, which needs to be supplemented.

Response 7: Thanks for your suggestion. We modified Figure 10 as you suggested..

Point 8: Section 4.2 of the article divides the image data set into training set, test set and validation set, but Figure 8 in Section 4.4 does not show the loss curve for validation, which needs to be supplemented.

Response 8: Thanks for your suggestion. We have supplemented Figure 8 and further explained it as you suggested.

Thanks and best regards.

Yours sincerely,

Jicheng Deng

Reviewer 2 Report

In this paper, an improved MobilenetV1-YOLOv4 algorithm is proposed for insulator defect detection, a practical industrial field problem. In addition, the authors introduce the scSE attention mechanism and depth-separable convolution into the overall network architecture for the purpose of improving monitoring accuracy and speed. Overall, this work sounds interesting and does address some of the problems observed by the authors in the industrial field. However, in terms of the current research work and the quality of the manuscript, I believe that publication in Entropy (ISSN 1099-4300) is still inadequate. The paper still leaves room for improvement in terms of writing and methodology, with the main issues being.

Comment 1: In the introduction, the authors provide a detailed description of the research background to this work. A brief review of the current research progress in the field is also provided. However, there are still some problems with the authors' logical relationships in the introductory section. The authors devote a large amount of space (Ref. 10 to Ref. 31) to the current dominant one-stage target detection algorithm, which in fact is not helpful for the description of the problem. Instead of reviewing each article in detail, the authors should summarize them and condense the specific scientific questions.

Comment 2: The author mentions “Although both two-stage and one-stage target detection algorithms have achieved great success in object recognition, due to the limitations of storage space and power consumption, many scholars have begun to study a lightweight model with reduced complexity and improved detection speed that can be deployed on tiny devices.” in the paper. But the review of the relevant literature that immediately follows is irrelevant to this description. The author needs to consider using a more rigorous logical relationship to explain the background of this study.

Comment 3: The grammar in the paper needs to be further revised. There are too many long sentences in the whole article. They are a little bloated and difficult to read. Give a simple example: “Literature [10] proposed an improved YOLOv5 insulator breakage detection algorithm, which firstly integrated the attention mechanism ECA-NET (Efficient Channel Attention) into the backbone feature extraction layer to improve the information interaction between channels, And secondly utilized bidirectional features to increase the promotion of small targets in the feature fusion layer to reduce the loss of small target information, and finally adopted the Soft NMS algorithm to reduce the original candidate frames to improve the detection accuracy of overlapping insulators.” In fact, there are many such phenomena in the whole article.

Comment 4: There is a small problem with the title of subsection 3.2. It should rather be Depthwise Separable convolutional module rather than Neural Network.

Comment 5: What do the Experiment and result analysis in '4 Experiment and result analysis' refer to respectively? I have seen more of an introduction to the web only.

Comment 6: In subsection 4.2 the author focuses on highlighting how to set labels on images, as well as describing the whole process in detail. This is useless. Because these steps are all too familiar to the workers involved. In fact, these steps should not be considered pre-processing of images either.

Comment 7: The author gives Figure 8, but I can't see any useful information in it. The author also just gives this simple diagram without any explanation.

Comment 8: The letterhead names in Tables 2, 3, 4 should be amended. More valuable information needs to be given.

Comment 9: The authors' discussion of the experimental results is too superficial, as these can be obtained directly from the table. It would have been more appropriate for the authors to analyze the reasons behind the occurrence of these phenomena.

Comment 10: It is recommended that the multiple plots in Figure 9 be plotted together to make it easier to make a quantitative assessment.

Comment 11: The authors should consider including some comparative experiments. Firstly, ablation experiments and secondly, comparison experiments with current state-of-the-art methods.

Comment 12: The authors present the advantages of the proposed method, so does it have some limitations? This is crucial for the development of the field.

Author Response

Response to Reviewer 2 Comments

In this paper, an improved MobilenetV1-YOLOv4 algorithm is proposed for insulator defect detection, a practical industrial field problem. In addition, the authors introduce the scSE attention mechanism and depth-separable convolution into the overall network architecture for the purpose of improving monitoring accuracy and speed. Overall, this work sounds interesting and does address some of the problems observed by the authors in the industrial field. However, in terms of the current research work and the quality of the manuscript, I believe that publication in Entropy (ISSN 1099-4300) is still inadequate. The paper still leaves room for improvement in terms of writing and methodology, with the main issues being.

Response :Thanks for your comment.

Point 1: In the introduction, the authors provide a detailed description of the research background to this work. A brief review of the current research progress in the field is also provided. However, there are still some problems with the authors' logical relationships in the introductory section. The authors devote a large amount of space (Ref. 10 to Ref. 31) to the current dominant one-stage target detection algorithm, which in fact is not helpful for the description of the problem. Instead of reviewing each article in detail, the authors should summarize them and condense the specific scientific questions.

Response 1: Thanks for your comment. We have made corrections in the introduction of the paper as you suggested.

Point 2: The author mentions “Although both two-stage and one-stage target detection algorithms have achieved great success in object recognition, due to the limitations of storage space and power consumption, many scholars have begun to study a lightweight model with reduced complexity and improved detection speed that can be deployed on tiny devices.” in the paper. But the review of the relevant literature that immediately follows is irrelevant to this description. The author needs to consider using a more rigorous logical relationship to explain the background of this study.

Response 2: Thank you for your advice. According to your suggestions, we have adjusted the introduction and cited the relevant papers [27-32] on lightweight models, which are more rigorous in the literature.

Point 3: The grammar in the paper needs to be further revised. There are too many long sentences in the whole article. They are a little bloated and difficult to read. Give a simple example: “Literature [10] proposed an improved YOLOv5 insulator breakage detection algorithm, which firstly integrated the attention mechanism ECA-NET (Efficient Channel Attention) into the backbone feature extraction layer to improve the information interaction between channels, And secondly utilized bidirectional features to increase the promotion of small targets in the feature fusion layer to reduce the loss of small target information, and finally adopted the Soft NMS algorithm to reduce the original candidate frames to improve the detection accuracy of overlapping insulators.” In fact, there are many such phenomena in the whole article.

Response 3: Thanks for your suggestion. We have adjusted some sentences in the paper as you suggested.

Point 4: There is a small problem with the title of subsection 3.2. It should rather be Depthwise Separable convolutional module rather than Neural Network.

Response 4: Thanks for your comment. We have changed the title of subsection 3.2 as you suggested.

Point 5: What do the Experiment and result analysis in '4 Experiment and result analysis' refer to respectively? I have seen more of an introduction to the web only.

Response 5: Thanks for your comment. Since section 4 is mainly platform construction and training process, the title has been changed. The experiment and result analysis are described in Section 5.

Point 6: In subsection 4.2 the author focuses on highlighting how to set labels on images, as well as describing the whole process in detail. This is useless. Because these steps are all too familiar to the workers involved. In fact, these steps should not be considered pre-processing of images either.

Response 6: Thanks for your suggestion. We have made changes in section 4.2 of the paper as you suggested.

Point 7: The author gives Figure 8, but I can't see any useful information in it. The author also just gives this simple diagram without any explanation.

Response 7: Thanks for your suggestion. We have supplemented the content of Figure 8 as you suggested.

Point 8: The letterhead names in Tables 2, 3, 4 should be amended. More valuable information needs to be given.

Response 8: Thanks for your comment. According to your suggestion,we have changed the names of Tables 2, 3 and 4 to provide more valuable information.

Point 9: The authors' discussion of the experimental results is too superficial, as these can be obtained directly from the table. It would have been more appropriate for the authors to analyze the reasons behind the occurrence of these phenomena.

Response 9: Thank you for your comment. We supplemented the discussion of the experimental results according to your suggestion. 14-15 pages in the paper

Point 10: It is recommended that the multiple plots in Figure 9 be plotted together to make it easier to make a quantitative assessment.

Response 10: Thanks for your suggestion. According to your suggestion, we draw several figures in Figure 9 together and correspond to the experiment

Point 11: The authors should consider including some comparative experiments. Firstly, ablation experiments and secondly, comparison experiments with current state-of-the-art methods.

Response 11: Thanks for your suggestion. According to your suggestion, we conducted some comparative experiments in Table 5 and compared it with YOLOv5.

Point 12: The authors present the advantages of the proposed method, so does it have some limitations? This is crucial for the development of the field.

Response 12: Thanks for your comment. We summarized its limitations on page 18 of the paper as you suggested.

Thanks and best regards.

Yours sincerely,

Jicheng Deng

Round 2

Reviewer 1 Report

The authors have answered some of my queries. However, the author response document is too simple. The authors must explain how they revise the manuscript and where is the revision in the text.

In addition, the revised manuscript removes some important and relevant references compared to the manuscript v1, e.g. [13], [15], [27]. I don't think it is appropriate to delete these references, because they are directly related to the topic of this manuscript. The authors must add these references to the Introduction and improve the background and research status.

Author Response

Response to Reviewer 1 Comments

Point 1: The authors have answered some of my queries. However, the author response document is too simple. The authors must explain how they revise the manuscript and where is the revision in the text. In addition, the revised manuscript removes some important and relevant references compared to the manuscript v1, e.g. [13], [15], [27]. I don't think it is appropriate to delete these references, because they are directly related to the topic of this manuscript. The authors must add these references to the Introduction and improve the background and research status.

Response 1: Thanks for your comment. For question 1 of the first round of the review report:” In section 3 of this paper, the author mentions "the DBRCBR module contains two kinds of convolutions, depthwise convolution and point convolution, to form depthwise separable convolution", while the "DBRCBR" described in Figure 2 only contains "depthwise convolution" and does not involve " depthwise separable convolution".” we added the operation process of point convolution on page 5 of the paper. The details are as follows: ” Conv2D in DBRCBR module is point convolution, and it is actually 1 × 1 Convolution. Its function is to freely change the number of output channels. Next, it is to perform channel fusion on the feature map of depth convolution output. After the convolution operations of depth convolution and point convolution, this is the operation process of depth separable convolution.” The relationship between depth convolution, point convolution and depth separable convolution is explained

For question 2 of the first round of the review report:” In section 4.4 of this paper, the author uses non-maximum suppression algorithm to obtain the prediction frame, but the parameter settings related to the non-maximum suppression algorithm are not given and need to be supplemented.” We give the details on page 11 of the text. The details are as follows:” The essence of non-maximum suppression method is to search local maxima and sup-press nonmax elements. First, set the confidence threshold of the target box. The threshold set in this paper is 0.5. Then, arrange the list of candidate boxes in descend-ing order according to the confidence. In the identified target categories, select and re-tain the bounding box A with the highest confidence, and then calculate the IoU of bounding box A and the remaining boxes B. If the IoU value is greater than the thresh-old, remove B. Repeat this step until the iteration of a target class is completed and the required target box is finally output.” The specific steps of the non maximum suppression algorithm are described.

For question 3 of the first round of the review report:” In this paper, the scSE attention mechanism is added to the PANet network, and it is best to explain the impact of its added position and number on the model detection results, and provide a theoretical basis.” We give the details on page 13 and page 15 of the text. The details are as follows:” This is because scSE can automatically learn the effective features of an image from both image space and feature channel, suppress useless redundant features, and better retain image edge information. Finally, more useful features are obtained through fea-ture splicing and convolution. The improved algorithm can identify insulator defects better. In Table 5, First, only add scSE attention mechanism (2 scSEs) after upsampling results. Sec-ondly, add scSE (3 scSEs) after preliminary feature extraction. This paper argues that embedding scSE attention modules into different parts of the network will produce different experimental results. The semantic information of the feature map initially extracted is not rich, but it still retains the medium and shallow texture information and contour information of the target in the feature map. This information is very im-portant for target detection. After the initial extraction of the three dimensions, em-bedding the scSE attention module can better enhance the spatial features and channel features of the target in the feature map. In the PANet structure, its feature map shows richer semantic features, larger receptive field and smaller feature map scale. The scSE attention module can no longer effectively distinguish important spatial and channel features from the highly fused small-scale feature map. In the experiment, we can know that with the gradual increase of scSE, the accu-racy gradually increases, but the detection speed becomes smaller and smaller. Because in the same network, the network parameters are also increased.” This specifically describes the impact of the number of scSEs and the location of addition on the model

For question 4 of the first round of the review report:” This paper proposes an insulator defect detection method based on the improved MobilenetV1-YOLOv4, which introduces the MobilenetV1 backbone feature extraction network, depthwise separable convolution and scSE attention mechanism, thus to greatly reduces the number of parameters, but the mAP value is increased by 4.4 %, whether the model has sufficient generalization ability for other related datasets.” We give the details on page 19. The details are as follows:” With the emergence of a new target detection algorithm, the algorithm proposed in this paper needs to be further optimized. At present, it will take some time to complete the hardware deployment. In the future, we will consider applying this model to other detections to achieve the real-time performance and generalization ability of this algo-rithm.”

For question 5 of the first round of the review report:” In section 5 of this paper, the author provides different models of P-R curves, but no relevant comparative analysis is carried out, which needs to be supplemented.” We give the details on page 16. The details are as follows:” In the P-R curve, P represents precision and R represents recall. With recall as the abscissa axis and precision as the ordinate axis, the P-R curve can intuitively display the overall accuracy and recall of the classification algorithm. We can illustrate the superiority of the algorithm by comparing the area size under the P-R curve. It can be seen from the figure (a-c) that the area of the improved MobilenetV1-YOLOv4 curve is large, indicating that its detection accuracy is good. The green line represents YOLOv5, which occupies the largest area due to its highest accuracy.”

For question 6 of the first round of the review report:” In section 5 of this paper, the author gives the results of the improved MobilenetV1-YOLOv4 detection of insulators, and does not give the detection results of other related models. It is recommended to supplement different models to predict the comparison results of the same test image.” We give the details on page 16. The details are as follows:” Figure 10 is a chart of insulator defect prediction results. No matter whether the insulator defects in the figure are large or small, they can be detected well without missing detection. And in each picture, the confidence level of the improved algorithm is higher than YOLOv4, indicating that the detection accuracy of this algorithm is higher. When the defect is at the edge of the image, the improved YOLOv4 detection effect is better.”

For question 7 of the first round of the review report:” Some image samples in the data set are given in Figure 7 in Section 4.2 of this paper, but they are all single images with a single insulator defect. Whether there are multiple insulator defects in a single image, which needs to be supplemented.” In Figure 10, we show the detection of several insulator defects

For question 8 of the first round of the review report:” Section 4.2 of the article divides the image data set into training set, test set and validation set, but Figure 8 in Section 4.4 does not show the loss curve for validation, which needs to be supplemented.” We have supplemented Figure 8 and further explained it as you suggested. The details are as follows: “As shown in Figure (8), the loss function of training set and verification set has a significant downward trend. After 200 times of training, the loss function gradually converges to a fixed value. After reaching 250 times, loss function is basically un-changed. This indicates that the improved network has high convergence speed and the trainings work well.”

Thanks for your suggestion. These references [13] [15] [27] and other references have been added to the introduction. Respectively corresponding to the literature [27] [28] [29] in the text, and newly added literature [30]. The details are as follows: “Literature [27] proposed a new insulator defect detection algorithm using deep learn-ing and morphological detection, which adopted the residual network to extract the morphological features of insulators, and then the image segmentation pixel clustering method to establish a mathematical model of insulator defects. The detection accuracy of this method is good, but the network parameters are more complicated. Literature [28] proposed an insulator string detection method based on improved YOLOv5, which adopted the EIOU loss function and the AFK-MC2 anchor point generation method to detect insulators. Literature [29] proposed an insulator defect detection based on the improved lightweight YOLOv4, reaching a detection accuracy of 93.81% and a detection speed of 53 frames per second. Literature [30] proposed an improved insulator detection algorithm based on YOLOvX, which adopted an improved Siou loss function to speed up model convergence, and were embedded with the ECA attention mechanism. Research results showed that the detection accuracy of this method reached 97.18%, and the detection speed 71 frames per second. And in the following part, we will quote the lightweight network related literature. The specific contents are as follows: Traditional CNN has a large demand for memory and computation, which makes it unable to run on mobile devices and embedded devices. MobileNet [31-33] was pro-posed by the Google team in 2017, focusing on lightweight CNN networks in mobile terminals or embedded devices. It used Depth Convolution in the network to reduce the amount of computation and parameters. Using the reciprocal residual structure in the network can reduce the memory consumption during reasoning. Wang [34] pro-posed CSPNet lightweight network in 2019. By integrating gradient changes into the feature map from beginning to end, CSPNet can reduce the amount of computation and ensure accuracy. This method can reduce the amount of calculation and improve the running speed of the model without reducing the accuracy of the model. It can be deployed on the mobile end. In 2020, Huawei proposed a new lightweight network GhostNet [35]. We know that the redundancy of feature map is very important, so we have designed a Ghost Module module. It realized the operation with less computation to generate these redundant feature maps. And deploy the network to the mobile ter-minal. In order to make the algorithm model smaller, we can apply it to insulator de-tection. ” This leads to the network structure of this article, making the article more reasonable.

Thanks and best regards.

Yours sincerely,

Jicheng Deng

Reviewer 2 Report

The author gave serious and professional answers to the questions in the first round of review. They really answered my question. Therefore, I think the current version can be considered for acceptance.

Author Response

Response to Reviewer 2 Comments

Point 1: The author gave serious and professional answers to the questions in the first round of review. They really answered my question. Therefore, I think the current version can be considered for acceptance.

Response1 :Thanks for your comment.

Thanks and best regards.

Yours sincerely,

Jicheng Deng

Round 3

Reviewer 1 Report

The authors have answered all of my quries. I think this manuscript can be accepted in present form.